# Visualizing MiRNA Regulation of Apoptosis for Investigating the Feasibility of MiRNA-Targeted Therapy Using a Fluorescent Nanoprobe

**DOI:** 10.3390/pharmaceutics14071349

**Published:** 2022-06-25

**Authors:** Mingyao Ren, Zhe Chen, Chuandong Ge, Wei Hu, Jing Xu, Limin Yang, Mingming Luan, Nianxing Wang

**Affiliations:** 1Shandong Provincial Key Laboratory of Molecular Engineering, School of Chemistry and Chemical Engineering, Qilu University of Technology (Shandong Academy of Sciences), Jinan 250353, China; rmyao1997@163.com (M.R.); 17861405811@163.com (Z.C.); gechuandong2021@163.com (C.G.); weihu@qlu.edu.cn (W.H.); xujing@qlu.edu.cn (J.X.); 2College of Chemistry and Pharmaceutical Sciences, Qingdao Agricultural University, Qingdao 266109, China; yanglimin@qau.edu.cn

**Keywords:** fluorescent nanoprobe, miRNA, apoptosis, cell imaging, real-time in situ detection

## Abstract

MiRNA-targeted therapy is an active research field in precision cancer therapy. Studying the effect of miRNA expression changes on apoptosis is important for evaluating miRNA-targeted therapy and realizing personalized precision therapy for cancer patients. Here, a new fluorescent nanoprobe was designed for the simultaneous imaging of miRNA-21 and apoptotic protein caspase-3 in cancer cells by using gold nanoparticles as the core and polydopamine as the shell. Confocal imaging indicated that the nanoprobe could be successfully applied for in situ monitoring of miRNA regulation of apoptosis. This design strategy is critical for investigating the feasibility of miRNA-targeted therapy, screening new anti-cancer drugs targeting miRNA, and developing personalized treatment plans.

## 1. Introduction

Cancer is a devastating and highly fatal disease that leads to cancer diagnosis and treatment, inspiring many research fronts in the creation of various detection and therapy platforms. Precision medicine offers personalized and tailored treatment according to the patient’s genes (e.g., miRNA, mRNA, DNA), lifestyle (e.g., diet, smoking, drinking), and living environment [1]. Such personalized treatment can offer a high cure rate, fewer side effects, and increased treatment precision. The importance of personalized treatment has risen to the national strategic level, and miRNA-based targeted therapy is currently an active research field in precision cancer therapy.

MiRNAs are small and non-coding RNAs with a nucleotide sequence of 20–24 nt [2]. Abnormal expression of miRNAs is closely related to the occurrence and development of various cancers [3,4]. MiRNAs play significant roles in important processes, including cancer cell apoptosis, invasion, and metastasis, by regulating target genes. Studies have shown that apoptotic genes can be regulated to induce cell apoptosis by enhancing or inhibiting the expression of specific miRNAs. In this regard, miRNAs are candidates for achieving precise cancer therapy [5,6,7]. Apoptosis is autonomous cellular programmed death that can remove functionally impaired cells. Real-time monitoring of apoptosis is critical in drug efficacy and prognosis evaluation. Different miRNAs play distinctive roles as oncogenic factors or tumor suppressors in the human body [8,9,10,11,12,13]. For example, miRNA-21 is overexpressed as an oncogene in various cancers, including pancreatic cancer, ovarian cancer, lung cancer, and colorectal cancer; let-7 is a tumor suppressor gene down-regulated in the cancers described above [14]. Thus, it is necessary to understand the effects of changes in miRNA expression on apoptosis when studying miRNA-targeted precision cancer therapy.

Current methods for detecting intracellular miRNAs are mainly based on molecular biology techniques such as Northern blotting, microarray analysis, and the quantitative reverse transcription–polymerase chain reaction (qRT–PCR) [15,16]. Methods for the study of apoptosis include fluorescence microscopy, electron microscopy, immunohistochemical analysis, caspase-activity assays, and Western blotting (WB) [17,18,19,20,21]. Although these approaches are commonly applied to detect miRNA and apoptotic proteins, most require a process of cell lysis or fixing, which cannot accurately reflect the real-time changes in miRNA expression and apoptosis in living cells. In addition, these approaches also suffer from cumbersome and time-consuming experimental operations and require many specimens. Therefore, it is particularly essential to develop a real-time, in situ detection method for tracking intracellular miRNA and apoptosis and visualizing the effect of changes in miRNA expression on apoptosis.

Recently, fluorescent nanoprobes have been successfully applied for the detection of miRNA or apoptosis-related markers in living cells due to a great many advantages, including non-invasiveness, visual identification, and extremely small consumption [22,23,24,25,26,27,28,29,30,31,32,33]. However, the fluorescence imaging methods mostly realize the single detection of miRNA or apoptosis in living cells, which cannot in situ monitor the effect of miRNA expression changes on apoptosis.

Here, we construct a new fluorescent nanoprobe for the simultaneous detection of miRNA-21 and caspase-3 in living cells and the in situ visualization of miRNA regulation of apoptosis. The nanoprobe possesses core–shell architecture, with gold nanoparticles (AuNPs) as a core and polydopamine (PDA) as a shell. Owing to the abundant catechol and amino groups on the PDA surface, the gold–PDA core–shell nanoparticles (Au@PDA NPs) were gradually functionalized with Cy5-labeled single-strand DNA (ssDNA) and FITC-labeled oligopeptide, step-by-step, via π-π and electrostatic interactions [34]. The resultant Au@PDA–ssDNA–peptide NPs (termed the nanoprobe) can specifically identify miRNA-21 and caspase-3 (Figure 1A). According to a previous study, we hypothesized that the caveolae-mediated pathway mainly contributes to the endocytosis pathways of the nanoprobe. The nanoprobe can partly escape from endosomes/lysosomes into the cytoplasm [35,36]. There are many amino groups on the PDA shell; this may be responsible for allowing the nanoprobe to escape from endosomes/lysosomes into the cytoplasm by the proton sponge effect [37]. The ssDNA modified on the nanoprobe has a specific sequence that can recognize specific targets (miRNA-21). Additionally, the oligopeptide modified on the nanoprobe can be cleaved specifically by caspase-3. When the nanoprobe enters the cytoplasm, in the absence of targets, the AuNPs and PDA can effectively quench the fluorescence of the labeled dyes with ssDNA and oligopeptides [38,39]. The ssDNA will become double-stranded with the target chain and the oligopeptide will be cleaved by caspase-3 when the nanoprobe encounters the corresponding miRNA-21 and caspase-3 targets, thereby dissociating the fluorophores from the Au@PDA NPs and restoring the fluorescence (Figure 1B). The designed nanoprobe can be used to visually monitor miRNA regulation of apoptosis in living cells.

## 2. Materials and Methods

### 2.1. Preparation of Au@PDA NPs

AuNPs were synthesized according to methods reported previously [40]; 0.01% HAuCl_4_ aqueous solution was boiled, with stirring, and then 2 mL of 1% sodium citrate aqueous solution was rapidly added and stirred vigorously. After 15 min, the heating was stopped, and the reaction solution was allowed to cool to room temperature. The resulting AuNPs were stored at 4 °C for future use.

To coat the PDA shell with the AuNPs, 1 mg of dopamine hydrochloride was first dissolved in 8 mL of 10 mM Tris-HCl (pH 8.5), and 8 mL of 0.125 nM AuNP solution was rapidly stirred with the above dopamine solution for 1 h at room temperature. The resulting Au@PDA NPs were then purified via repeated centrifugation (13,000 rpm, 10 min), and the purified Au@PDA NPs were re-dispersed in ultrapure water for future use.

### 2.2. Fluorescence Quenching

Cy5-labeled ssDNA (150 nM) was added to various concentrations of Au@PDA NPs (10 mM HEPES, 5 mM CaCl_2_, pH 7.4) to obtain ssDNA-modified Au@PDA NPs (Au@PDA–ssDNA NPs). The fluorescence intensity of Cy5 at 667 nm was recorded under an excitation wavelength at 648 nm to determine the optimal amount of Au@PDA NPs. FITC-labeled oligopeptide (250 nM) was added to different concentrations of Au@PDA-ssDNA NP solutions (10 mM HEPES, 5 mM CaCl_2_, pH 7.4) to obtain peptide-modified Au@PDA–ssDNA NPs (Au@PDA–ssDNA–peptide NPs). The fluorescence intensity of FITC at 520 nm was collected at 480 nm excitation to determine the optimal amount of Au@PDA–ssDNA NPs. The experiments were repeated three times.

### 2.3. Preparation of Fluorescent Nanoprobe

The ssDNA21 solution (150 nM 10 mM HEPES, pH 7.4) was first added to the AuNP@PDA NP solution (5 nM) and incubated for 3 h at room temperature. Then, the oligopeptide solution (250 nM) was added to the above solution, followed by continuous incubation for 3 h. The resulting mixture was centrifuged and purified before dispersing it in a 10 mM HEPES buffer solution (5 mM CaCl_2_, pH 7.4) to obtain the nanoprobe modified with ssDNA and oligopeptide.

### 2.4. Fluorescence Response of the Nanoprobe

In order to detect miRNA-21, different concentrations of the perfectly matched miRNA-21 target strands (0, 10, 20, 30, 40, 50, 75, 125, 250, 750, and 1500 nM) were added to the nanoprobe solution (5 nM) and incubated for 1 h at 37 °C. The fluorescence intensity of Cy5-labeled ssDNA was collected with λex/λem = 648 nm/667 nm. For caspase-3 detection, various concentrations of caspase-3 (0, 1, 2, 3, and 4 units) were added to the nanoprobe solution (5 nM) and incubated for 4 h at 37 °C. The fluorescence intensity of FITC was recorded at λex/λem = 480 nm/520 nm. The experiments were repeated three times.

### 2.5. Kinetics

The fully matched miRNA-21 target strands (1000 nM) were added to the nanoprobe solution (5 nM) and incubated with increasing time (0, 5, 10, 15, 20, 30, 40, 50, 60 min) at 37 °C. The recombinant caspase-3 (4 units) was added to the nanoprobe solution (5 nM) and incubated at different times (0, 0.5, 1, 2, 3, 4 h). The fluorescence intensity of Cy5-labeled ssDNA was collected at λex/λem = 648 nm/667 nm, and the fluorescence intensity of FITC-labeled oligopeptide was recorded at λex/λem = 480 nm/520 nm. The experiments were repeated three times.

### 2.6. Specificity Experiments

To investigate the specificity of the nanoprobe toward miRNA-21, perfectly complementary miRNA-21 targets and other targets were examined. The concentration for each was 500 nM.

After incubating with the nanoprobe (5 nM) for 1 h at 37 °C, the fluorescence of Cy5 was excited at 648 nm and recorded at 667 nm. To detect the specificity of the nanoprobe toward caspase-3, caspase-3 (4 units), and other interfering substances, including hemoglobin (HGB, 4 μg/mL), bovine serum albumin (BSA, 4 μg/mL), glutathione (GSH, 1 mM), glucose (Glu, 20 mM), CaCl_2_ (2 mM), and NaCl (100 mM), were examined. The fluorescence intensity of FITC was excited at 480 nm and recorded at 520 nm after incubation for 4 h at 37 °C. All experiments were repeated three times.

### 2.7. MTT Assay

To evaluate biological toxicity of the nanoprobe, a tetrazolium-based colorimetric MTT assay was performed. A549 and Hela cells were seeded into 96-well microtiter plates (1 × 10^6^ cells/well), respectively. The culture was kept in a 5% CO_2_/95% air incubator at 37 °C for 24 h. Then the initial medium was removed, and the cells were cultured with different concentrations of Au@PDA and Au@PDA–ssDNA–peptide (1, 2, 4, 5, 8 nM) for 24 h, respectively. The MTT solution (5 mg/mL in PBS, 10 μL) was added to each well and further incubated for 4 h. After discarding the remaining MTT solution, 150 μL DMSO was added to dissolve the purple formazan. The absorbance at 490 nm was recorded with a Multiskan FC. The experiments were repeated three times.

### 2.8. Fluorescence Confocal Imaging

In the experiments for imaging miRNA-21 and caspase-3 in living cells treated with different drugs, Hela and A549 cells were chosen and divided into three groups in parallel. One group was treated with lipopolysaccharide (LPS, 10 μg/mL) for 12 h, and another group was treated with staurosporine (STS, 0.1 μM) for 12 h. The untreated group served as the control. After three groups of cells were incubated with the nanoprobe (5 nM) for 4 h, confocal laser scanning microscopy (CLSM) was performed. Red channels (miRNA-21) and green channels (caspase-3) were excited at 633 and 488 nm, and the fluorescence intensities were collected between 660–700 and 500–600 nm, respectively.

In the experiments for visualizing miRNA regulation of apoptosis in cells treated with transfection reagents, Hela and A549 cells were seeded in three confocal dishes to achieve a cell density of 30–50% at the time of transfection, respectively. One group of cells was transfected with miRNA-21 mimics (100 pmol) and another group of cells was transfected with miRNA-21 inhibitors (500 pmol) for 48 h according to the manufacturer’s instructions. The group of cells without transfection served as the control. The three groups of cells were cultured with the fluorescent nanoprobe (5 nM) for 4 h and then monitored by CLSM after washing the cells with PBS.

## 3. Results and Discussion

### 3.1. Preparation and Characterization of the Nanoprobe

We first synthesized AuNPs as the core of the fluorescent nanoprobe and then coated the PDA shell on the surface of the AuNPs through the in situ polymerization of dopamine. Transmission electron microscopy (TEM) results showed that the AuNPs had good morphology, dispersibility, and uniform size (18 ± 3 nm; Figure 1A). The thickness of the coated PDA shell was 10 ± 2 nm. The size of the Au@PDA NPs was approximately 32 ± 3 nm (Figure 1B). When ssDNA and oligopeptide were used to modify the Au@PDA NPs, the particle diameter increased slightly to about 35 ± 3 nm (Figure 1C). In addition, ultraviolet-visible absorption spectroscopy (UV–vis) analysis results showed that the maximum absorption peak shifted from 530 to 540 nm after PDA shell coating on the AuNPs. Additionally, the absorption peak further red-shifted slightly after ssDNA and oligopeptide modifications of the Au@PDA NPs (Figure 1D). There are several reasons that cause this phenomenon. The increased particle size of the nanoprobe is an important reason. Moreover, the maximum peak is red-shifted due to the higher amount of amino groups on the PDA and carboxyl groups on the oligopeptide, which increases the polarity of the solution [41]. Thus, this indicates that the PDA was successfully coated on the surface of the AuNPs. Dynamic light scattering (DLS) measurements indicate that the diameter of the Au@PDA NPs was mostly 32.7 ± 5 nm (Appendix A), which is consistent with the TEM data. The diameter of a few nanoprobes ranged from 30–90 nm; we suspect that the nanoprobe had unevenly dispersed in the solution and aggregated. The modification of the nanoprobe was further confirmed by zeta potential analysis. Figure 1E showed that the potential of AuNPs was −7.9 mV; the potential decreased to −13.2 mV after PDA coating. When ssDNA and oligopeptide were used to modify the Au@PDA NPs, the potential increased to −5.1 mV. These results confirm that the fluorescent nanoprobe was successfully prepared.

To achieve the best recognizing performance, the optimized concentration of Au@PDA NPs was then determined. Different concentrations of Au@PDA NPs were added to the 150 nM ssDNA solution to successfully prepare the fluorescence-quenched Au@PDA–ssDNA NPs. Figure 2A,C show that the 3.5 nM Au@PDA NPs completely quenched the fluorescence of Cy5 dye-conjugated ssDNA. Then, different concentrations of Au@PDA–ssDNA NPs with the quenching of Cy5 fluorescence were added to the 250 nM oligopeptide solution. The fluorescence of FITC-labeled oligopeptides was completely quenched when the concentration of Au@PDA NPs reached 5 nM (Figure 2B,D). Hence, the Au@PDA NPs at 5 nM were selected for the following experiments.

### 3.2. In Vitro Studies of the Nanoprobe

To study the feasibility of simultaneous detection of miRNA-21 and caspase-3 by the nanoprobe, we next investigated the response of ssDNA to a fully complementary target strand as well as the response of the oligopeptide to the corresponding caspase-3 protein. The kinetic results indicated that the nanoprobe could respond rapidly to the perfectly complementary paired miRNA-21 target strand within 10 min and reach a plateau within 60 min (Figure 3A and Appendix A). The fluorescence response of the nanoprobe to caspase-3 also increased gradually with time and reached a plateau after 3 h (Figure 3B and Appendix A). In the response experiment of the nanoprobe to the target, with a gradual increase in the perfectly matched target-21 concentration (0–1500 nM), the fluorescent signal of Cy5, labeled on ssDNA, also gradually increased (Figure 4A,C). Similarly, with a gradual increase in the concentration of caspase-3 protein (0–4 unit), the fluorescence intensity of the FITC labeled on the oligopeptide increased gradually (Figure 4B,D). The fluorescence intensity was linearly correlated to the concentration of caspase-3. These results indicate that the recovery of the fluorescence signal is due to the reaction of the nanoprobe with the corresponding DNA target strand and the caspase-3 protein. The fluorescent nanoprobe can achieve simultaneous detection of miRNA-21 and caspase-3.

To investigate whether the nanoprobe can specifically detect miRNA-21 and caspase-3 proteins, we analyzed the effects of other interfering substances on the nanoprobe. The mismatched target-21 strand and other targets (including target-221 and Ki-67) all showed marginal fluorescence changes compared with the control (Figure 3C). In contrast, the fully matched target-21 strand presented remarkable fluorescence responses. Similar results were detected when the fluorescent nanoprobe met the caspase-3 protein and other interfering substances (Figure 3D). We observed no obvious fluorescence recovery for any interfering substance except for caspase-7. Although caspase-7 can also recognize oligopeptides and result in the recovery of fluorescence, caspase-3 still performed about 3-fold better than caspase-7. The result is consistent with previous studies [42,43]. Therefore, the results show that the nanoprobe can selectively detect miRNA-21 and caspase-3 proteins with high specificity.

### 3.3. MTT Experiment

A549 and Hela cells were used to measure the cytotoxicity of the nanoprobe. As shown in Figure 5, the viability of both cell types remained over 90% after incubating the nanoprobe with cells for 24 h, which indicated that the nanoprobe had low cytotoxicity and good biocompatibility. Moreover, the two types of cells had a higher survival rate after being incubated with Au@PDA–ssDNA–peptide NPs than Au@PDA NPs. This is due to the better biocompatibility of the nanoprobe after the modification of ssDNA and peptides. We then set up three parallel groups with the two types of cells to examine whether their cell morphology changed after incubation with the nanoprobe. One group was treated with Au@PDA NPs, and the other group was treated with Au@PDA–ssDNA–peptide NPs. Additionally, the untreated group served as the control. The morphology of the cells treated Au@PDA NPs and Au@PDA–ssDNA–peptide NPs remained almost unchanged compared with the control group (Appendix A). The results confirm that the nanoprobe can be applied for intracellular detection and imaging.

### 3.4. Intracellular Imaging of MiRNA-21 and Caspase-3

To investigate the ability of the nanoprobe to simultaneously detect miRNA-21 and caspase-3 expression and activity levels in living cells, A549 cells and Hela cells were first chosen. It has been reported that the expression of miRNA-21 is particularly high in cancer cells and the expression and activity of caspase-3 are higher during cancer cell programmed death. We treated the two cell lines with LPS and STS. LPS can stimulate cells to produce an inflammatory response, and miRNA-21 expression is increased [44]. STS can induce cell apoptosis, and caspase-3 expression is increased [45]. Hela cells were divided in parallel into three groups. One group was treated with LPS; another group was treated with STS. The untreated group served as the control. After being incubated with the nanoprobe for 4 h, three groups of Hela cells were monitored by CLSM to measure the red fluorescent signal of Cy5, which was used to monitor miRNA-21, and the green fluorescence of FITC, which was used to monitor caspase-3 protein (Figure 6A). The red fluorescence was significantly enhanced and the green fluorescence was not changed in cells treated with LPS compared to the untreated cells. The green fluorescence was significantly increased and the red fluorescence was weakened in cells treated with STS compared with the control group. Grayscale values of fluorescence intensities obtained from CLSM further verified the expression changes of miRNA-21 and caspase-3 (Appendix A).

RT-PCR and WB methods were also applied to detect the relative expression of miRNA-21 and caspase-3 in Hela cells. As shown in Figure 6B,C, the results of RT-PCR and WB maintained consistency with the CLSM results. Meanwhile, the CLSM, PCR, and WB experiments in A549 cells were also performed, and all the results were similar to those obtained in Hela cells (Figure 6D–F). The results imply that the fluorescent nanoprobe can simultaneously detect miRNA-21 and caspase-3 protein in living cells.

To explore the capability of the nanoprobe to monitor miRNA regulation of apoptosis in living cells, Hela cells and A549 cells transfected with miRNA-21 mimics or inhibitors were further tested for the impact of miRNA expression on caspase-3. Compared to Hela cells without transfection treatment, the red fluorescence was significantly enhanced and green fluorescence was not observed in the cells transfected with miRNA-21 mimics, while the cells transfected with miRNA-21 inhibitors showed decreased red fluorescence and enhanced green fluorescence (Figure 7A). Grayscale values of fluorescence intensities are shown in Appendix A. The CLSM results were further verified using RT-PCR (Figure 7B) and WB (Figure 7C), which indicated that the reduced expression of miRNA-21 caused apoptosis in Hela cells and that the nanoprobe could successfully visualize miRNA regulation of apoptosis in living cells. A549 cells behaved similarly to Hela cells in CLSM, PCR, and WB (Figure 7D–F). These results suggest that miRNA-21 could be a potential therapeutic target candidate for Hela cells and A549 cells.

## 4. Conclusions

In summary, we have designed a new type of fluorescent nanoprobe that enables the simultaneous detection and imaging of miRNA and apoptotic markers in living cells. The nanoprobe can overcome the limitations of molecular biology methods and simultaneously monitor the expression levels of intracellular miRNA-21 and caspase-3 with high specificity and good biocompatibility. Fluorescence imaging results showed that the nanoprobe could be successfully applied for real-time monitoring of the effect of miRNA-21 expression changes on apoptosis by dual-color imaging. This design strategy provides a new method for in situ monitoring of miRNA-regulated apoptosis in cancer cells that is beneficial for assessing the feasibility of miRNA-targeted therapy. We anticipate that the nanoprobe could be a promising tool for evaluating anti-cancer drugs targeting miRNA and personalized therapy.

## Data Availability

The authors confirm the data supporting the findings of this study.

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
