# Peer review of "Visualizing MiRNA Regulation of Apoptosis for Investigating the Feasibility of MiRNA-Targeted Therapy Using a Fluorescent Nanoprobe"

_pharmaceutics, 2022, doi:10.3390/pharmaceutics14071349_

Round 1
Reviewer 1 Report
The present manuscript describes a novel Au nanoparticle-based fluorescent probe for simultaneous two-channel detection of target miRNA and activity of caspase-3. The authors demonstrated efficient readout for both targets for experiments in vitro and in live cultured cells. The work is interesting, well written and illustrated. However, some points need to be addressed:
Major points
1. In Scheme 1, nanoparticles directly enters into cytosol through the plasma membrane. Obviously, this is a multistep process, which should include attachment on the cell surface, endocytosis, and exit from vesicles to cytosol. It is of critical importance for the method, as nanoparticles can contact with miRNA and caspase in cytosol only. Please describe possible mechanisms of nanoparticle entrance based on literature and/or own data.
2. Since the authors have access to electron microscopy (see Fig. 1), ideally, they should show TEM of cells with nanoparticles in cytosol (not in intracellular vesicles).
3. In Figures 6 and 7, quantification of fluorescence changes in green and red channels should be shown (e.g., as histograms).
Minor points
4. Page 6, lines 208-211: “The kinetic results indicated that the nanoprobe could respond rapidly to the perfectly complementary paired miRNA-21 target strand within 10 min and reach equilibrium within 60 min (Fig. 3A and S2A). The fluorescence response of the nanoprobe to caspase-3 also increased gradually with time and reached equilibrium after 3 h (Fig. 3B and S2B).” I think the word "plateau" should be used here, not "equilibrium".
5. Figure 7 legend: “Scale bar is 50 μm”. In fact, panels D have scale bar 25 μm. Please make it consistent.
Reviewer 2 Report
From the viewpoint of chemistry, the MS represents the sensing procedure based on the quenching of the organic dyes fluorescence by the plasmon resonance AuNPs with further fluorescence restoration through the stripping of the dye-labelled oligopeptides and ssDNA-strands, in turn, derived from their specific and selective binding with miRNA-21 and caspase-3 protein. The work deserves acceptance after the highlighting of the several issues.
1. The construction of such nanoarchitecture is based on non-covalent layer-by-layer deposition. Indeed, the deposition of polydopamine seems to be the key stage for an efficiency of the further adsorption of the dye-labelled oligopeptide and ssDNA molecules. The authors represent three arguments for the efficient deposition of PDA: (1) electrokinetic potential of the citrate-stabilized AuNPs changes from -7.9 mV to -13 mV; (2) the UV-Vis band arisen from the plasmonic resonance of AuNPs exhibit 10 nm red-shift; (3) the additional shell appears at the TEM images of the AuNPs after the PDA deposition. No problems with the third argument, it is clear. But I wonder why electroneutral PDA deposition generates more negative exterior charging than the anionic citrates stabilizing the as-prepared AuNPs? The band’s shifting indicates some changes at the Au/water interface, but how does it correlate with PDA-deposition?
2. The kinetic of the fluorescent response of the NPs to miRNA-21 and caspase-3 protein is rather low from the viewpoint of small molecular sensing. It means that the stripping event is slow, since the interaction between the biomolecules should be rather quick, isn’t it? In this assumption the stripping event should be in the competition with layer-by-layer deposition of miRNA-21 and caspase-3 protein onto the as-prepared NPs. It will be great is the authors can discuss this issue.
3. To be available to wide circle of readers the authors should designate some “biological” abbreviations, such as WB, LPS, STS and some others.
Reviewer 3 Report
The manuscript by Mingyao Ren et al. describes the development, characterization and application of a Au@PDE-ssDNA-peptide nanoprobe for imaging miRNA21 and caspase-3 activity in cells. While the approach is interesting for such analyses, it is not a total novelty to the field that these nanoprobes can be used for such approaches, however, the combination of the two different nanoprobes is new. Nevertheless, I have several comments which should be addressed by the authors in order to significantly improve the quality of the manuscript.
Introduction:
The authors should include fluorescence microscopy and caspase-activity assays to the methods for apoptosis visualization. Subsequently, especially with fluorescence microscopy using e.g. FRET-based caspase activity reporters, the authors should modify the following sentence, as only some methods require cell fixation and lysis.
The authors MUST include the following background information in the introduction:
1) https://doi.org/10.1039/C8AN02145G
2) https://doi.org/10.1007/s00604-019-3573-8
3) https://doi.org/10.1039/D1RA00154J
These publications demonstrate that respective AuNPs were already existing and it is not a total novelty to the field that caspase-3 and miRNA21 can be visualized using (Au)NPs.
Results and Discussion:
The authors should include a TEM image of Au@PDA-ssDNA-peptide coated NPs. What is the size of these NPs? Why not showing images of this modified NPs?
In Figure 1c, no text describes/ discusses the Au@PDA-ssDNA-peptide coating. Additionally, it seems that the absorbance peak is further shifted if the ssDNDA-peptide is present. Please add a respective paragraph in the text.
Line 207: remove “.”
Please demonstrate/ test reversibility for miRNA21 interaction.
While the specificity of the ssDNA for its target – miRNA21 – is well controlled, it is not for the peptide-caspase-3 interaction. Better controls should be performed, including the testing of different caspases – especially caspase 7. Additionally, other major caspases (e.g. caspase-8/9/6/2/10/…) should be investigated, in order to ensure caspase-3 specificity.
Additionally – does the nanoprobe report caspase-3 even despite it is not activated? Usually, caspase-3 should not be enzymatically active without activation (i.e. prior cleavage by initiator caspases). The authors should investigate this issue more deeply.
In line 232-233 the authors claim that the nanoprobe can simultaneously detect miRNA21 and caspase-3 activity. Nevertheless, none of the experiments performed so far shows a combination of both. The authors should combine different of their “testing-interfering substances” and analyse both, FL Intensity at 648/667 corresponding to the ssDNA AND 480/520 corresponding to the peptide to ensure specificity of the two wavelengths and no cross-interference.
Did the authors statistically analyse their data from Figure 5?
In Figure 5: Why Au@PDA-DNA21-peptide and not Au@PDA-ssDNA-peptide?
Figure 5: The authors should include respective bright-field images of cells either non-treated with a nanoprobe or treated with the two different Au NPs. Does cell morphology change in response to NP treatment?
Lines 249-250: Rather “expression and activity levels” than only “expression levels”
Lines 252: The nanoprobe might not assess caspase 3 expression levels but activity?
Figure 6: The authors should include respective fluorescence quantifications of “Caspase-3” and miRNA-21 for both cell lines. Additionally – in the images it rather seems that it is an “either” – “or” signal – either miRNA-21 is observed OR caspase-3 – Hence: What happens under these conditions if LPS AND STS are applied simultaneously? The authors should investigate. Again, in the panels statistical comparisons are missing..
Figure 7: Similarly to Figure 6, quantifications of the fluorescence intensities as well as statistics are missing..
Throughout the manuscript, the authors should not only describe the results, but also discuss respective outcomes. E.g.
1) Why does the size of the Au@PDA range from ~ 30 – 90 nm as seen in Figure S1?
2) Why does the absorbance peak seem to shift upon modifying Au@PDA to the Au@PDA-ssDNA-peptide?
3) Throughout the manuscript, it is not clear why binding of miRNA-21 to the ssDNA on the Au@PDA-ssDNA-peptide yields an increase in fluorescence. While this is obvious for the peptide due to enzymatic cleavage, it is not clear for the reader why binding of the specific miRNA target mediates an increase in fluorescence. Please clarify.
4) The authors assessed the times required to reach equilibrium in fluorescence increase upon incubating their Au@PDA-ssDNA-peptide with either caspase-3 or miRNA21. As such effect/ time is tightly dependent on the concentration of both, nanoprobe and “substrate”, the authors should include respective information.
5) Throughout the manuscript, the authors should include information about the reversibility of the nanoprobe in response to caspase-3 or miRNA21 mediated activation. While caspase-3 activation might not be reversible, what about miRNA21 interaction? Does the method represent an endpoint measurement?
6) Why does the Au@PDA-ssDNA-peptide nanoprobe seem to reduce the cell viability less than Au@PDA? Please discuss.
Throughout the manuscript, the authors should include the number of replicates!
Round 2
Reviewer 1 Report
I have no more comments.
Author Response
Thank you for your decision.
Reviewer 3 Report
The authors have addressed most of my comments.
The only aspects missing are:
8) and 14) I am a bit worried about why the authors do not show the simultaneous administration of Caspase-3 AND miRNA21 in vitro, OR the application of STS AND LPS on living cells. Concerning 8), the authors changed the text accodring to their experiments performed. Nevertheless, it would be interesting/ important to see simultaneous incubation of the AuNPs with caspase-3 AND miRNA-21 (in vitro) or STS and LPS on cells... Is there any cross-interference of the wavelengths/ activation patterns, which the authors do not want to show?
12/13) AuNP needs to be cleaved by caspase-3. Hence I would suggest, as mentioned, "expression AND activity".
18) I cannot find the statement in page 3, lines 83-90
